# Advances in Research on the Involvement of Selenium in Regulating Plant Ecosystems

**DOI:** 10.3390/plants11202712

**Published:** 2022-10-14

**Authors:** Wei Chao, Shen Rao, Qiangwen Chen, Weiwei Zhang, Yongling Liao, Jiabao Ye, Shuiyuan Cheng, Xiaoyan Yang, Feng Xu

**Affiliations:** 1College of Horticulture and Gardening, Yangtze University, Jingzhou 434025, China; 2Engineering Research Center of Ecology and Agricultural Use of Wetland of Ministry of Education, Yangtze University, Jingzhou 434025, China; 3Hubei Key Laboratory of Waterlogging Disaster and Agricultural Use of Wetland, Yangtze University, Jingzhou 434025, China; 4National R&D Center for Se-Rich Agricultural Products Processing, School of Modern Industry for Selenium Science and Engineering, Wuhan Polytechnic University, Wuhan 430023, China; 5Henry Fok School of Biology and Agricultural, Shaoguan University, Shaoguan 512005, China

**Keywords:** selenium, plant, allelopathy, ecosystem

## Abstract

Selenium is an essential trace element which plays an important role in human immune regulation and disease prevention. Plants absorb inorganic selenium (selenite or selenate) from the soil and convert it into various organic selenides (such as seleno amino acids, selenoproteins, and volatile selenides) via the sulfur metabolic pathway. These organic selenides are important sources of dietary selenium supplementation for humans. Organoselenides can promote plant growth, improve nutritional quality, and play an important regulatory function in plant ecosystems. The release of selenium-containing compounds into the soil by Se hyperaccumulators can promote the growth of Se accumulators but inhibit the growth and distribution of non-Se accumulators. Volatile selenides with specific odors have a deterrent effect on herbivores, reducing their feeding on plants. Soil microorganisms can effectively promote the uptake and transformation of selenium in plants, and organic selenides in plants can improve the tolerance of plants to pathogenic bacteria. Although selenium is not an essential trace element for plants, the right amount of selenium has important physiological and ecological benefits for them. This review summarizes recent research related to the functions of selenium in plant ecosystems to provide a deeper understanding of the significance of this element in plant physiology and ecosystems and to serve as a theoretical basis and technical support for the full exploitation and rational application of the ecological functions of selenium-accumulating plants.

## 1. Introduction

Selenium is an essential trace mineral nutrient for humans and animals and has important functions in enhancing immunity, scavenging free radicals in the body, and preventing cardiovascular diseases [1]. Selenium deficiency poses a great risk to human health, triggering macrosomia, immune deficiency, impaired fertility, Keshan disease, and Kashin–Beck disease [2]. However, excessive Se intake also causes selenosis, the main symptoms of which are respiratory distress, vomiting, abdominal pain, and diarrhea [3]. Selenium is widely but heterogeneously distributed in the natural environment, and soil total Se content is significantly correlated with the soil parent material and climatic conditions. The average total Se content of soil around the world ranges from 0.1 mg·kg^−1^ to 2.0 mg·kg^−1^, and the median concentration in Chinese soils is 0.219 mg·kg^−^^1^ [4]. In the typically Se-rich areas in Enshi Tujia and Miao Autonomous Prefecture, Hubei Province, the Se content in the topsoil is 0.84 ± 1.39 mg·kg^−1^ [5]. Soil Se deficiency zones are widespread across the globe, and currently approximately 0.5–1.0 billion people are at risk of Se deficiency [6]. For example, most of China’s land is a Se-deficient zone [2]. The low intake of micronutrients in the Chinese diet causes the problem of “hidden hunger.” Therefore, it is necessary to supplement Se in a scientific and reasonable way to maintain the health of the people.

Plants can convert absorbed inorganic Se into selenoprotein, an excellent carrier and human food source for Se supplementation, and other forms of organic Se. Selenium-rich crops are being rapidly developing as a mainstay in Se-rich functional agriculture to meet the demand for Se supplementation. At present, the regions of Hubei Enshi and Shaanxi Ankang in China rely on the advantage of the local soil being a Se-rich resource to strongly support the development of Se-rich agriculture and produce Se-enriched grains and horticultural crops, such as rice, wheat, potatos, and tea [2,7,8,9]. Other nonnatural Se-rich areas primarily use exogenous Se fertilizers to improve the Se content of crops to meet consumer demand [10,11]. The development of Se-rich functional agriculture is of great significance in meeting the health needs of the nation. Hence, Se-rich agricultural products and deeply processed functional foods have a broad developmental prospect.

Through plant uptake, translocation, and metabolism into other forms of Se-containing substances, soil-based Se is accumulated in the plant body or released into the environment in the form of volatiles. These Se-containing metabolites produce specific ecological effects that affect ecological relationships among plants, animals, and microorganisms. Basing on recent studies on the relationship between Se and plants, this paper explores the role of Se in plant ecosystems and provides theoretical references for research on the ecological restoration, cultivation, and production of Se-rich crops and the ecological significance of Se accumulation in plants.

## 2. Selenium Uptake and Metabolism by Plants

Plants absorb Se in the form of selenite, selenate, and organic Se mainly through their roots [12,13]. Soil is the main source of Se uptake by plants. The form of the Se in soil is affected by the soil clay content, redox potential, and pH [14,15]. Selenium and sulfur are elements of the chalcogen and have similar chemical properties. As shown in Figure 1, selenate is mainly transported by sulfate transport proteins and metabolized along the sulfur metabolic pathway in plants [16,17]. Selenate is converted to 5′-selenadenosine phosphate by ATP sulfatase in chloroplasts and then transformed into selenite through the reduction of adenosine 5′-adenosine phosphate reductase. Selenocysteine (SeCys) is then synthesized by selenite and acetylserine (OAS) under the catalysis of cysteine synthetase (CSase). SeCys has three main metabolic pathways in plants. The first is being broken down to alanine (Ala) and elemental Se (Se^0^) by selenocysteine lyase. The second is the formation of methylselenocysteine (MeSeCys) by selenocysteine methyltransferase. MeSeCys can be further synthesized into volatile dimethyl diselenide (DMDSe), which helps plants to reduce their Se content. The third is the conversion of SeCys into selenomethionine (SeMet); in chloroplasts, SeCys is first converted by cysthioether γ-synthase and cystathionine β-lyase and then catalyzed by methionine synthase to form SeMet. In some higher plants, SeMet can be produced by methionine methyltransferase (MMT). Under the action of MMT, the SeMet from higher plants can produce methylselenomethionine (MeSeMet), which in turn can further form volatile dimethyl selenide (DMSe). In addition, plants can store SeCys and SeMet in the chloroplast or cytoplasm. Selenium primarily accumulates in plants in the form of seleno amino acids and selenoproteins and also as various Se-containing compounds, such as Se polysaccharide [13,18,19].

Selenite can be taken up by the root system through phosphate transporter proteins, such as rice *OsPT2* [20], and transported to other parts of the plant via the xylem, where selenite is rapidly converted into organoselenium compounds [21,22,23]. There are other ways for selenite to enter plants; rice, for example, can absorb selenite through aquaporins and the Si influx transporter OsNIP2;1 [24,25]. Amino acid transporters can also take up these organic Se forms [19]. The rate of Se uptake by plants is influenced by several factors, such as plant species, soil Se concentration, and Se forms. It was reported that when exposed to 5.0 μM exogenous Se, the absorption rate of selenite, SeMet, and SeCys in *Brassica napus* was 1.5, 40, and 2 times than that of selenate, respectively. In addition, the absorption rate of selenite, SeMet, and SeCys in *Triticum turgidum* is 5, 100, and 20 times than that of selenite [19].

According to the Se content, plants in their natural habitats can be classified into Se hyperaccumulators (>1000 mg·kg^−1^ DW), Se accumulators (100–1000 mg·kg^−1^ DW), and non-Se accumulators (<100 mg·kg^−1^ DW) [26]. Therefore, the Se-accumulation ability of different species of plants is different. The Se accumulation mechanism of hyperaccumulator plants is an important scientific issue that is worthy of our attention, but the current research is not extensive enough to clarify the underlying mechanism [27]. Previous studies showed that *Stanleya pinnata* sequestered Se in localized epidermal cell clusters along leaf margins and tips [28]. It has been found that the Se hyperaccumulator *Cardamine violifolia* has Se-enriched regions in the tips of shoot and roots and at the apical meristems, while the non-Se accumulators *Cardamine pratensis* tends to concentrate small amount of Se in the vasculature [29]. In addition, Se hyperaccumulators have higher levels of antioxidants, which may help these plants prevent Se-associated oxidative stress and tolerate high concentrations of Se [28].

Recently, important progress has been made in the identification and functional characterization of genes related to the Se accumulation mechanism of plants. It was reported that sulfate transporter (SULTR) genes in Se hyperaccumulators (*S. pinnata* and *C. violifolia*) usually have a high expression level, which may contribute to the efficient uptake and transport of inorganic Se forms in the soil by Se hyperaccumulators [30,31]. Furthermore, researchers have identified SMT enzymes from a variety of Se hyperaccumulators and non-Se accumulators of the *Astragalus* species, but only SMT isoforms in Se hyperaccumulators (*Astragalus bisulcatus*) have the activity to catalyze the synthesis of MeSeCys [32], which indicates that SMT plays an indispensable role in Se hyperaccumulators in context of accumulating and tolerating high Se concentrations. SMT genes have been widely used in transgenic research to enhance Se accumulation and tolerance in non-Se accumulators. For example, the SMT gene isolated from *A. bisulcatus* was expressed in *Arabidopsis thaliana*, which significantly increased the accumulation of MeSeCys and γ-glutamyl-methylselenoselenoic acid in *A. thaliana* [33]. The expression of the AbSMT gene in Indian mustard significantly increased the content of MeSeCys [34], and MeSeCys can also be further converted into volatile DMDSe [35]. In summary, the current understanding of the Se accumulation mechanism of hyperaccumulators is insufficient, and more in-depth research is urgently needed to reveal the specific molecular mechanism.

Inorganic Se is converted by plants to organic Se or other Se-containing compounds through transformation and metabolic processes and then are accumulated or excreted in the plant [12,26,36]. Selenium is not an essential nutrient in plants. Higher plants have lost all selenoprotein genes during evolution and are unable to synthesize selenoproteins. Meanwhile, humans, animals, and some thallophytes retain the ability to synthesize selenoproteins [37]. On the one hand, appropriate levels of Se can promote plant growth and development and improve plant resistance to biotic or abiotic stresses, among its other important functions [38,39,40,41]. On the other hand, high concentration of Se can be toxic to plants due to the excessive misassembly of SeCys and SeMet into proteins, causing changes in protein spatial structure, affecting protein activity, and forming physiological disorders that can be toxic to plant cells [42,43].

Plants synthesize a variety of selenides through physiological processes such as the absorption, transformation, and metabolism of Se, which can regulate their growth and development. Meanwhile, the Se compounds secreted by plants can change the forms and distribution of Se in the adjacent soil [44]. Selenium compounds can be used as allelochemicals to affect the relationship between plants and ecosystems through allelopathy [45,46]. In this process, plants or microorganisms use some allelochemicals to produce beneficial or harmful effects on other surrounding organisms. This phenomenon is common in plant ecosystems. For example, the germination rate of *A. thaliana* seeds planted in the rhizosphere soil of *A. bisulcatus* was significantly reduced [47]. Another study found that canopy coverage near Se hyperaccumulators decreased by 7% [48]. At present, we have a relatively clear understanding of the absorption and transformation pathway of selenate in plants, but there is insufficient research on the metabolic mechanism for selenite and organic Se, which is the important direction that needs more in-depth research.

## 3. Selenium Affects Plant-Plant Interactions

Selenium hyperaccumulators can negatively affect the growth and distribution of non-Se accumulators in their vicinity by altering the Se content of the local soil to obtain a large amount of sunlight, water, and other resources for their reproductive survival [46]. This phenomenon is one of the Se-polymerizing dynamics of Se hyperaccumulators. The Se content in *A. bisulcatus* root domain soil can reach 71–103 mg·kg^−1^ DW, which is three–five times higher than that in the inter-root environment of non-Se accumulators [47]. Se-sensitive *A. thaliana* grown in the root domain soil of Se hyperaccumulators had a significantly lower germination rate and dry weight compared with those grown in the root domain soil of non-Se accumulators [47]. These findings suggest that plants may use their Se-polymerizing ability to improve their ecological advantage.

Mehdawi et al. [49] indicated that Se hyperaccumulators promote the growth of nearby Se-tolerant plants. The Se content of soil around the Se hyperaccumulator *S. pinnata* was approximately 7–13 times higher than that in the soil around non-Se accumulators. The Se content in the leaves of *Artemisia ludoviciana* and *Symphyotrichum ericoides* growing naturally near the Se hyperaccumulator *S. pinnata* in Colorado, USA was 10–20 times that when they are at a distance from Se hyperaccumulators. In addition, the plant heigh increased by two–three times, and statistically the plants had more leaves when they were grown near Se hyperaccumulators. This finding indicates that Se hyperaccumulators significantly promote the growth of *A. ludoviciana* and *S. ericoides* in their vicinity [49]. The above study showed that Se hyperaccumulators have more pronounced ecological effects compared with their surrounding Se accumulators and non-Se accumulators.

Plants release organic Se into the soil in the form of apoplast and root secretions, which increase the organic Se content into the soil and are readily absorbed and utilized by plants [25]. The Se content and organic Se content in soil are proportional to the Se content in the surrounding plants [49]. Reynolds et al. studied three different Se-enriched areas in the United States and found high Se content in the soil in the Pine Ridge Natural Area, Cathy Fromme Prairie Natural Area, and Coyote Ridge Natural Area close by to the Se hyperaccumulators *S. pinnata* and *A. bisulcatus* [48]. In addition, the Se content decreased rapidly beyond 50 cm of their roots. This finding was consistent with the canopy width of hyperaccumulators and indicated that hyperaccumulator plants have a concentration effect in terms of Se, that is, the high Se content in their tissues has a significant effect on their adjacent plant communities [48]. The high Se content of the soil under the plant canopy may be caused by the high Se content in the fallen leaves; root secretions may also play a role. In addition, hyperaccumulators can transfer Se from underground to surface soils through the root system or enrich Se to the central soil, forming a relatively high Se “enrichment zone” [49]. By comparing the Se content of soil within 3 m around 22 Se hyperaccumulators and 22 non-hyperaccumulators at the Pine Ridge Natural Area, the average Se content around hyperaccumulators was two times higher than that around non-hyperaccumulators. This result suggests that the Se content of soil is related to Se hyperaccumulators [48]. A 5% increase in bare land area, 7% increase in plant species richness, and 7% decrease in plant canopy cover were observed in the area within 1.5 m of Se hyperaccumulators compared with that of non-Se hyperaccumulators [48].

A possible cause for this is that the excessive Se content in the soil near hyperaccumulators restricts the growth of the surrounding plants and to some extent inhibits the growth of some plants that should occupy most ecological niches, allowing less competitive species to have access to resources, such as sunlight and water, for growth and reproduction. As a result, the characteristics of the surface vegetation are altered [44,50,51]. In a word, Se hyperaccumulators can collect, transform, and release Se in the ecological environment by an absorption and transformation progress. In turn, the above-mentioned process changes the Se speciation and distribution in soil, thus regulating the vegetation characteristics in the surrounding environment of Se hyperaccumulators.

## 4. Selenium Affects Plant-Animal Interactions

Selenium in soil is absorbed, transformed, and metabolized by plants to produce other Se-containing compounds, some of which are volatile, such as DMSe and DMDSe. These volatile selenides often have special odors that have a deterrent effect on herbivores, and some can even be detected by human olfaction [28]. Quinn et al. [52] found that Se in plant tissues and volatile Se compounds can reduce the foraging behavior of herbivores. In a field setting, *Cynomys ludovicianus* groundhogs tend to feed on plants with low Se content within the same plant species; this behavior was consistent with findings in relation to non-Se accumulators, such as *Brassica juncea*, versus Se hyperaccumulators, such as *S. pinnata*. *B. juncea* reduced the chance of feeding by *C. ludovicianus* groundhogs when the Se content reached 38 mg·kg^−1^ DW or above [52]. *C. ludovicianus* and some Orthoptera may rely on their olfactory perception of volatile Se compounds [52,53].

High levels of selenide can be a deterrent and toxic to animals [53]. Aphids (*Myzus persicae*) avoid eating *B. juncea* leaves with Se levels of up to 10 mg·kg^−1^ DW. Feeding aphids using Indian mustard leaves with a Se content of 1.5 mg·kg^−1^ DW caused Se poisoning and death [54]. When the Se concentration exceeded 0.4 mg·kg^−1^, the pupation rate of honeybee (*Apis mellifera*) larvae was significantly reduced, indicating that high Se concentrations have a negative effect on *A. mellifera* larvae [55]. The consumption of pollen and nectar with high Se contents reduced the pupation rate of *A. mellifera* larvae and impaired the normal physiological functions of honeybees. Long-term memory tests on honeybees revealed that feeding with sugar water containing 3 μL of sodium selenate at a concentration of 6 mg·L^−1^ decreased the memory of bees, indicating that exposure to excess Se causes bees to exhibit learning and memory deficits. This effect may reduce the ability of worker bees to gather resources and care for the colony [56]. When honeybees (*Apis mellifera* L.) were fed different concentrations of selenate and selenite (60–600 mg·L^−1^), they exhibited different degrees of oxidative stress [57]. Excess Se is harmful to animals, but Se deficiency decreases the ability of animals to synthesize antioxidant enzymes (such as SOD), which leads to enhanced lipid peroxidation activity and increased free radical levels and malondialdehyde content. Moderate amounts of supplemental Se can improve these symptoms [58]. The dietary inclusion of 0.32–0.36 mg·kg^−1^ sodium selenite was effective in enhancing the antioxidant capacity of Italian honeybee larvae and improving their pre-pupal weight and pupation rate [55]. These results indicate that excessive Se may damage the normal physiological function of animals.

Long-term natural selection has made some animals tolerant to Se, and this relative Se tolerance may help plants gain an ecological advantage in environments with high Se contents. The Se content accumulated in the flowers of the Se hyperaccumulator *S. pinnata* can reach 2323 mg·kg^−1^ DW, and the nectar can contain 244 mg·kg^−1^ FW [51]. The Se in *S. pinnata* pollen does not hinder the pollination behavior of honeybees, and 0.4–1.0 mg·kg^−1^ FW of Se was detected in adult bees that fed on these pollens [51]. To investigate whether high levels of Se affect bees’ pollen feeding on plants, Colin et al. placed high and low Se plants (treated with 80 μmoL and 0 μmoL Se, respectively) at 10 m from the colony. During the experiment, the positions of the plants were exchanged once [51]. The results showed that the bees did not avoid the pollen of plants with high Se contents, and no significant difference in the number of foraging sessions was observed between the high- and low Se pollens [51]. A similar finding was obtained in another study where the number of pollen pickings by bees did not decrease or increase after the Se treatment of radish (*Raphanus sativus*) [59].

Freeman et al. [60] found that a variety of *Plutella xylostella* could survive normally on plants with high Se contents. The Se content in larvae and adults was approximately 200 mg·kg ^−1^ DW, which is 10 times higher than that in normal *P. xylostella*. Comparative studies between Se-tolerant and Se-sensitive *P. xylostella* revealed that the former had no significant preference for high and low Se plants and the latter laid significantly fewer eggs on high Se plants than on low Se plants. The natural screening of Se-tolerant moths on high-Se plants, especially Se hyperaccumulators, possibly allowed this variant to tolerate Se at higher levels compared with other Se-sensitive moths, thus facilitating competition between Se-tolerant moths and other herbivores. The study also found that Se was mainly distributed in the abdomen of wild-living Se-resistant *P. xylostella*, and the Se content of parasitoids was also high, with the high Se content in their bodies likely protecting them from predators. Selenium-tolerant *P. xylostella* isolate Se in their hindgut or in deposits on the outside of their abdomen, suggesting that they may use the accumulated Se to reduce the risk of predation [60,61]. The above studies showed that pollinators did not take the initiative to avoid Se-containing plants, implying that high Se contents in plants may prompt pollinators to evolve Se tolerance. The research on the regulating function of Se between plants and pollinators is still insufficient, but the question of how insects evolve Se tolerance under the pressure of environmental selection with high Se contents is an important scientific issue that deserves our continuous attention.

## 5. Selenium Affects Plant-Microbe Interactions

Plant growth and development are closely related to the activity of microbes, which are distributed within the plant, on the surface, and between the roots. The relationship between microorganisms and plants may be mutually beneficial or neutral or may lead to diseases. Plants provide microorganisms with the nutrients required for their life activities, and microorganisms improve plant resistance and facilitate water and nutrient uptake by plants [62]. Microorganisms can assist plants in the uptake of Se from the environment. Inoculation with *Funneliformis mosseae* and *Glomus versiform* for eight weeks significantly promoted the uptake of selenate and selenite by the root systems of wheat (*Triticum aestivum*) seedlings which were planted in the sterilized soil sand mixture [63]. Indian mustard inoculated with rhizosphere bacteria exhibited an approximately fivefold increase in root Se content compared with the commonly cultivated plant [64]. Some microorganisms can influence the forms of Se in plants. Some endophytic bacteria isolated from plants can reduce sodium selenite to elemental Se in media containing sodium selenite [65]. The elemental Se in the Se hyperaccumulators *S. pinnata* and *A. bisulcatus* accounts for 35% of their total Se content [61]. However, an experiment involving growing sterilized seeds of these two plants in a greenhouse revealed that these two Se hyperaccumulators mainly accumulated organic Se, such as seleno amino acids, and did not accumulate elemental Se in their bodies [66]. Elemental Se is not soluble for plant cells and is excreted by the plant, so these endophytic bacteria can help the plant to reduce Se accumulation and enhance the host plant’s tolerance to Se.

Selenium in plants has a suppressive effect on some pathogenic microorganisms, and Se hyperaccumulators can enhance the disease resistance of plants. For example, Hanson et al. reported improved disease resistance in mustard seedlings treated with Se [67]. After sterile mustard seedlings grown on agar medium with or without Se were immersed in *Fusarium* sp. spore suspension, the Se-treated mustard seedlings (with a Se content of approximately 800 mg·kg^−1^ DW) and the control were infected by *Fusarium* sp. The fresh weight of the mustard seedlings without Se decreased by 14% after seven days, and that of mustard seedlings with Se increased by 1%. This result indicates that a certain level of Se in mustard seedlings can reduce the damage caused by the pathogenic fusarium on mustard. Therefore, enrichment with certain concentrations of Se enhances plants’ resistance to pathogenic bacterial infestation, and this effect may be one of the intrinsic motivations for Se enrichment in plants.

Plant root microorganisms isolated from soils with high Se contents are more tolerant to Se than those from low Se areas. An analysis of the Se tolerance of plant root microorganisms collected from four Se-enriched and one non-Se-enriched regions in Colorado and Wyoming revealed that plant root microorganisms from Se-enriched regions were not sensitive to 10 mg·L^−1^ sodium selenates in the medium. Meanwhile, the microorganisms from non-Se-enriched regions were highly sensitive to the same concentration of Se [68]. The greater Se tolerance of plant root microorganisms from Se-enriched areas suggests that they may have an adaptive advantage to Se-enriched areas. Given that the Se content in the roots of Se hyperaccumulators can reach 100 times that of non-Se accumulators, microbial populations growing in their vicinity face great evolutionary selection pressure [69]. In summary, microorganisms favor enhanced Se tolerance in host plants and resistance to environmental stress, and host plants promote Se tolerance in inter-rooted microorganisms to some extent. Plants release organic Se into the soil, the absorption and transformation of organic and inorganic Se by microorganisms is an important intermediary in the Se cycle. Therefore, the coevolution mechanism of plants and microorganisms in high Se content environments is an interesting scientific topic.

## 6. Conclusions and Outlook

Although Se is not an essential trace element for plants, it plays an important role in plant physiological and biochemical pathways and ecosystems. As shown in Figure 2, an appropriate amount of Se supplementation promotes plant growth and adaptation to the environment. However, high concentrations of Se inhibit plant growth and even produce toxicity. The uneven distribution of Se in soil and the difference in plant tolerance in terms of Se affects the diversity of associated species. Through the uptake, transformation, and release of Se, Se hyperaccumulators increase the Se content of soil in the root domain, promote the growth of Se-tolerant plants, inhibit the growth of Se-sensitive plants, increase the abundance of microbial species in the root zone, and enhance plant disease resistance. High levels of Se in plants can reduce foraging by herbivores and increase the Se content in pollinators, giving plants an ecological advantage in environments with high Se content. However, high concentrations of Se inhibit plant growth and even cause poisoning symptoms. Selenium hyperaccumulators can enhance the total Se content and organic Se content in the inter-root soil through root secretions and the apoplast, forming a small Se “enrichment zone”. Selenium can be enriched in the food chain, forming an ecosystem characterized by high Se content and promoting the evolution of plants, animals, and microorganisms toward Se tolerance within a certain range.

This paper has summarized the research on the important role of selenium in plant ecology in recent years. However, there remain many important issues that are unclear and need further study. Firstly, the Se-accumulation mechanism of Se hyperaccumulators is not completely known. *C. violifolia* is a typical Se hyperaccumulated plant which is only distributed in China and could be selected as an excellent experimental material to explore the Se-accumulation mechanism and reveal the differences between hyperaccumulators and non-hyperaccumulators. Additionally, the ecological function and significance of *C. violifolia* has been rarely studied and discussed. More in-depth research is urgently needed to explore the relationship and underlying mechanism between *C. violifolia* and plants, animals, and microorganisms in the inhabited environment. Moreover, plants and microorganisms are the crucial intermediaries responsible for the transportation and transformation of organic and inorganic Se in the natural cycle of Se. Plants and microorganisms have an inseparable relationship in adapting to inhabited environments with high Se contenta. Therefore, the co-evolution mechanism of Se accumulated between plants and microorganisms is an attractive scientific issue and deserves further exploration.

In conclusion, research on the function of Se in plant ecosystems is still in its infancy. Studying the content, forms, and effects of Se in different links of the ecological chain will help us understand the functions of this mineral in plant growth and development and related ecosystems more deeply, elucidate how Se-polymerizing plants affect the distribution, vegetation characteristics, and ecological effects of Se in soil, and provide a theoretical basis and scientific guidance for the land management planning, ecological restoration and application, and comprehensive development of Se-rich agriculture in Se-rich areas.

## Figures and Tables

**Figure 1 plants-11-02712-f001:**
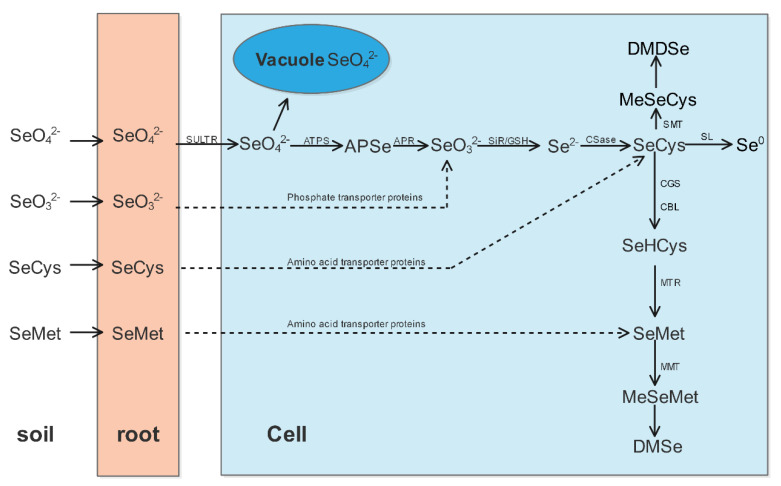
The main ways plants absorb and assimilate selenium. Note: SULTR, sulfate transporter; ATPS, ATP sulfurylase; APSe, adenosine 5′-phosphoselenate; APR, adenosine 5′-phosphosulfate reductase; SiR, sulfite reductase; GSH, glutathione; CSase, cysteine synthase; SMT, selenocysteine methyltransferase; SL, selenocysteine lyases; CGS, cystathionine-γ-synthase; CBL, cystathionine β-lyase; MTR, methionine synthase; MMT, methionine methyltransferase; SeCys, selenocysteine; SeMet, selenomethionine; SeHCys, Se homocysteine; SeMSeMet, methyl selenomethionine; SeMSeCys, methyl selenocysteine; DMSe, dimethyl selenide; DMDSe, dimethyl diselenide.

**Figure 2 plants-11-02712-f002:**
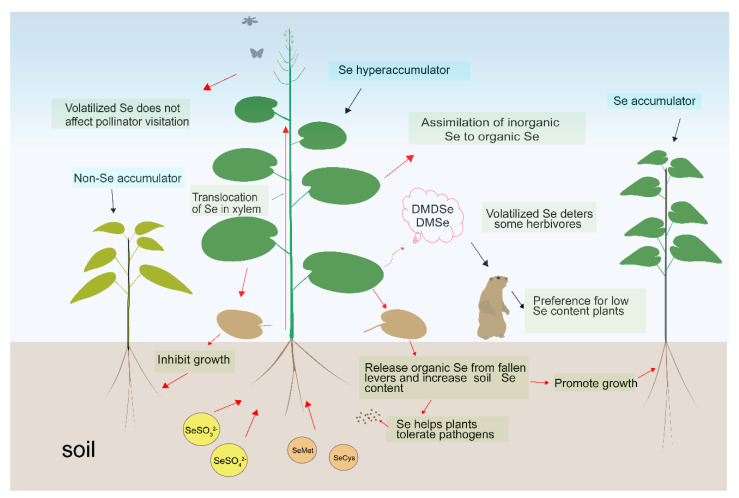
Pattern diagram of selenium participating in regulating plant ecosystem. Note: DMSe, dimethyl selenide; DMDSe, dimethyl diselenide; SeMet, selenomethionine; SeCys, selenocysteine.

## Data Availability

Not applicable.

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
