# Peer review of "Advances in Research on the Involvement of Selenium in Regulating Plant Ecosystems"

_plants, 2022, doi:10.3390/plants11202712_

Round 1
Reviewer 1 Report
This manuscript is organized well, and the figures are good. However, the English writing should be improved. There are other comments as well, see below:
1. Line42-43, “The average soil selenium content around the world ranges from 0.1 mg·kg-1 to 2.0 mg·kg-1 and the average selenium content in Chinese soils is 0.21 mg·kg-1”. Please add a reference here, and as I know, the average selenium content in Chinese soils is 0.29 mg·kg-1 , please check it.
2. Line 72-73, please add a reference.
3. Line 74, “genus sulfur” is not a terminology, chalcogen.
4. Line 98, Line 125, Line 184, and Line 165, please use “selenium forms” instead of “selenium morphology” here.
5. Line 142-144, “Selenium-sensitive Arabidopsis thaliana grown using the root domain soil of non-Se accumulators had significantly lower germination rate and dry weight compared with those grown using the root domain soil of Se hyperaccumulators.” Please check if the germination rate and dry weight were lower or higher here. If it was lower, the result cannot support your point here.
6. Line 152-154, “In addition, the biomass, plant height, and leaf number of these two plants statistically increased by 2–3 times when they were grown near non-selenium-polymerizing plants.” This result indicates that the non-Se accumulators rather than Se hyperaccumulators promote the growth of nearby 147 selenium-tolerant plants. Please check it.
7. Line 142 “Selenium-sensitive” and Line 148 “selenium-tolerant plants.”, please give the definitions of this, and distinguish this with Se hyperaccumulators, Se accumulators and non-Se accumulators. According to your fig. 2, if the selenium-sensitive plant is equal to non-Se accumulators and the selenium-tolerant plants is equal to Se accumulators? Please make it more clear.
8. Line 160, as I know, organic selenium is not readily absorbed 160 and utilized by plants.
9. Line 174-175, this sentence is confusing. what’s the result? and which 22 plants?
10. Line 267-268, “The elemental selenium in Se hyperaccumulators S. pinnata and A. bisulcatus accounts for 30% of their total selenium content.” Please check and add a reference here. I doubt about it, and I thought you also doubt it (because you give the negative reference then). So I am confusing.
11. In the full text, Se and selenium are mixed used. Please use abbreviations correctly
12. Fig. 2, please provide a high-resolution image.
13. Fig. 2 can be improved. If the red arrows mean the selenium flow? Why only organic selenium was observed in soil? The main selenium form in soil is not organic selenium.
Reviewer 2 Report
.......cardiovascular diseases (Reference?). LINE 36
...elements of genus sulfur???. LINE 74 (Not sure what this means)
[36]............LINE 74 (Period missing here after the reference)
....suggest that these 22 plants (LINE 74)... which 22 do you mean? elaborate with a statement about the 22 plant before this
Hladun et al. ....LINE 250. Check, it is incomplete, year missing and need formating to match rest of the references
......improve plant resistance diseases... LINE 258. Delete resistance
Se hyperaccumulator....LINE 312. Write Se in it complete name because it is teh beginning of a sentence
Reviewer 3 Report
The manuscript entitled "Advances in research on the involvement of selenium in regulating plant ecosystems" is a review article that briefly discusses the significance of selenium in human and animal nutrition, and overall selenium distribution in worldwide soils. The authors presented a more detailed overview on selenium uptake and metabolism in plants; the role of selenium in plant-plant, plant-animal and plant-microbe interactions.
This review comprises valuable articles that could contribute to our understanding and increase our interest in a beneficial element such as selenium and its potential role in the environment. However, before publication, several points require the authors' attention. The principal recommendations to the authors refer to the improvement of the content, critical overview and reference interpretation. The present review article could merit publication in Plants, section Phytochemistry, special issue Selenium Biology in Plants, only after an extensive revision.
Comments to authors:
Manuscript’s content revision
The manuscript comprises valuable content; however, this review article is missing the author’s personal view, critical opinion, and research suggestions on the presented topic. Even if there is a paragraph entitled Conclusions and outlook, it somehow missed fulfilling the content introduced by the title.
According to MDPI guidelines, herewith I cite “Reviews offer a comprehensive analysis of the existing literature within a field of study, identifying current gaps or problems. They should be critical and constructive and provide recommendations for future research. No new, unpublished data should be presented. The structure can include an Abstract, Keywords, Introduction, Relevant Sections, Discussion, Conclusions, and Future Directions, with a suggested minimum word count of 4000 words.”
Some articles presented in the manuscript were not available in English. When cited in review articles, included manuscripts should pose at least an English abstract. Herewith I leave the Editors and authors to decide whether the suggestion is plausible or not.
The manuscript comprises valuable content, but some parts require authors’ attention and minor revision:
1. I would recommend to revise sentences indicated in the attached PDF document (grey color). Within the PDF document some additional comments are provided.
2. Selenium uptake and metabolism in plants are explained in Section 2. However, the provided information missed pointing out the peculiars and differences in the metabolism and translocation of both Se-accumulators (for example line 81) and non-accumulators. When applicable, multiple metabolic paths should be presented by schematic diagrams along with detailed explanations within the manuscript's body.
3. Intake of selenite is enabled through phosphate transporter proteins, as stated in the article. In addition, there are other types of proteins, like aquaporins, with similar functions and increasing research interest.
References revision
All references and reported results should be checked even if not highlighted or indicated during the revision. I would recommend to review and compare the cited works with the written content throughout the manuscript. A few examples are indicated in the attached PDF document (light blue color). Within the PDF document some additional comments are provided.
1. I would recommend the authors to verify all the references, as some are missing (lines 147, 162, etc.).
2. I would recommend the authors to verify the citation accuracy of the reported results (lines 37, 39, 142, etc.).
Manuscript’s style revision
The manuscript should be carefully edited as I encountered some minor style inconsistencies. All of these errors won't be highlighted, therefore the manuscript needs to be revised in overall. A few examples are indicated in the attached PDF document (purple or light blue (references) color). Within the PDF document some additional comments are provided.
Here are listed some examples:
1. Scientific names are predominantly used when indicating the plant/animal species at the first mention, while only the abbreviation is listed later. This approach is adequate; however, I would suggest verifying the consistency of the adopted style (lines 201, 205, etc.).
2. References should be written according to MDPI’s style guidance.
Figures revision
Figures are great additions to review articles. Abbreviations adopted in Figures should be clearly described. Also, authors are encouraged to include cited references within the Figures’ caption title if this attends the journal's guidelines. Comments are indicated in the attached PDF document (red color).
1. Figure 1: In my opinion, Se-accumulators and non-accumulators should be separately indicated. The content of this Figure should be revised (indicated within the PDF document).
2. Figure 2: When it comes to the design, I have some concerns. The content of this Figure should be revised (indicated within the PDF document).
Concluding remarks
The presented comments aim to improve the quality of the manuscript. I consider the submitted review article of great potential value and significance for the field once it is revised.

Reviewer 4 Report
I find this review very interesting, allthough there are few very similar reviewes published previously. It would be great to improve your paper with more graphical displays and to discuss deeper about every topic/subtopic in your manuscript.
Round 2
Reviewer 1 Report
The manuscript has been reviewed well. I suggest to accept it.
Author Response
Thank you for your kind comments and affirmations.Reviewer 3 Report
The presented form of the article is significantly improved. I would kindly ask the authors to provide more details on two previously listed points.
Point 1. I expressed one of my main concerns and comments at the beginning of the review. The authors missed addressing it within the revised version or in the comments. In my opinion, this subject is significant as it highlights the MDPI’s guidelines for Review papers. The pointed statement is listed below. I leave it to the Editorial board to decide if this concern is plausible or not.
The manuscript comprises valuable content; however, this review article is missing the author’s personal view, critical opinion, and research suggestions on the presented topic. Even if there is a paragraph entitled Conclusions and outlook, it somehow missed fulfilling the content introduced by the title.
According to MDPI guidelines, herewith I cite “Reviews offer a comprehensive analysis of the existing literature within a field of study, identifying current gaps or problems. They should be critical and constructive and provide recommendations for future research. No new, unpublished data should be presented. The structure can include an Abstract, Keywords, Introduction, Relevant Sections, Discussion, Conclusions, and Future Directions, with a suggested minimum word count of 4000 words.”
Point 2. The authors sustained (Point 2; Point 7) that there is a small amount of literature regarding some written sections. I agree that additional studies to understand the essential paths in the Se metabolism are indispensable; however, fundamental knowledge on differences between Se metabolism in accumulators and non-accumulators is already available. There are a few recent Review papers* presenting the discussed matter to more extent.
White, P. J. (2018). Selenium metabolism in plants. Biochimica et Biophysica Acta (BBA)-General Subjects, 1862(11), 2333-2342.
Lima, L. W., Pilon-Smits, E. A., & Schiavon, M. (2018). Mechanisms of selenium hyperaccumulation in plants: A survey of molecular, biochemical and ecological cues. Biochimica et Biophysica Acta (BBA)-General Subjects, 1862(11), 2343-2353.
Reynolds, R. J. B., & Pilon-Smits, E. A. (2018). Plant selenium hyperaccumulation-Ecological effects and potential implications for selenium cycling and community structure. Biochimica et Biophysica Acta (BBA)-General Subjects, 1862(11), 2372-2382.
*Disclaimer: The reviewer is not the author of the listed papers and is unrelated to them. Indicated articles are listed solely to demonstrate the discussed case.
